# A Novel Semi-Soft Decision Scheme for Cooperative Spectrum Sensing in Cognitive Radio Networks

**DOI:** 10.3390/s19112522

**Published:** 2019-06-02

**Authors:** Yin Mi, Guangyue Lu, Yuxin Li, Zhiqiang Bao

**Affiliations:** 1Xi’an Institute of Optics and Precision Mechanics of CAS, University of Chinese Academy of Sciences, Xi’an 710119, China; miy@xupt.edu.cn; 2School of Communications and Information Engineering, Xi’an University of Posts and Telecommunications, Xi’an 710021, China; dreamyaurora@126.com (Y.L.); baozhiqiang@xupt.edu.cn (Z.B.)

**Keywords:** cognitive radio, cooperative spectrum sensing, soft decision, IoT

## Abstract

Spectrum sensing (SS) is an essential part of cognitive radio (CR) technology, and cooperative spectrum sensing (CSS) could efficiently improve the detection performance in environments with fading and shadowing effects, solving hidden terminal problems. Hard and Soft decision detection are usually employed at the fusion center (FC) to detect the presence or absence of the primary user (PU). However, soft decision detection achieves better sensing performance than hard decision detection at the expense of the local transmission band. In this paper, we propose a tradeoff scheme between the sensing performance and band cost. The sensing strategy is designed based on three modules. Firstly, a local detection module is used to detect the PU signal by energy detection (ED) and send decision results in terms of 1-bit or 2-bit information. Secondly, and most importantly, the FC estimates the received decision data through a data reconstruction module based on the statistical distribution such that the extra thresholds are not needed. Finally, a global decision module is in charge of fusing the estimated data and making a final decision. The results from a simulation show that the detection performance of the proposed scheme outperforms that of other algorithms. Moreover, savings on the transmission band cost can be made compared with soft decision detection.

## 1. Introduction

With the rapid development of the Internet of Things (IoT) [1] and wireless applications, especially for fifth generation (5G) communication systems and beyond [2], spectrum resources are becoming increasingly strained. Currently, the radio spectrum is used based on a fixed spectrum allocation policy. According to the Federal Communication Commission (FCC), the utilization of an allocated spectrum resource is about 15–85% below 3 GHz. In 2015, the FCC released a report that highlighted the urgent need to improve the spectrum efficiency [3]. Motivated by this, the cognitive radio (CR) technology was proposed to cope with the spectrum utilization [4].

In CR networks, the primary user (PU) has authority to use the spectrum bands for communication, but secondary users (SUs) can only opportunistically access the spectrum holes that are not occupied by the PU. As a result, they have to continuously detect the spectrum bands over a period of time. Then, the SUs need to instantly evacuate once the PU returns to the bands so that harmful interference can be reduced. In addition, another model allows the SUs to coexist with the PU in the licensed bands, but the PU’s predefined interference temperature must be maintained. This means that the SUs always transmit the information with a lower power to guarantee the PU’s quality-of-service (QoS), thereby greatly improving the spectrum efficiency. It is worth noting that spectrum sensing plays a vital role in CR [5,6,7]. In general, there are various path losses in an actual wireless communication scenario, such as multi-path fading effects, shadowing effects, and the near–far effect, which may degrade the spectrum sensing results. In order to ensure accurate spectrum sensing, cooperative spectrum sensing (CSS) has emerged as a solution in which multiple SUs collaboratively detect the spectrum bands. In the end, the SUs send local information to the fusion center (FC), which makes a final decision [8] through the reporting channel.

In general, there are two types of local information fusion. The first is Hard decision detection [9,10], in which each SU makes a local decision and reports the results via 1-bit data, where ‘1’ represents the presence of the PU and ‘0’ represents the PU’s absence. The final decision is made on the basis of specific fusion rules, such as the ‘OR’ rule, the ‘AND’ rule, and the ‘MAJORITY’ rule. The second is Soft-decision detection [10,11], in which each SU sends the sensing data to the FC and the final decision is made by constructing a global test statistic via fusing all of the local sensing information, which increases the local transmission band cost in contrast to 1-bit hard decision detection. Therefore, the tradeoff between the sensing performance and the band cost is a challenging issue.

In [12], each SU sends data on its decision to the FC through an unreliable reporting channel, and the n-out-of-*K* fusion rule is employed at the FC. The optimal number of cooperative SUs is analyzed concerning the minimum Bayes risk. A tri-threshold CSS based on weighing is proposed to improve the detection performance in [13], and obtains better detection performance than single and double-threshold detection under a low signal-to-noise ratio (SNR).

However, the final decision is made by the FC using a hard-decision rule. Soft decision detection enjoys a significant competitive advantage in sensing performance over hard-decision detection. Hence, researchers have been encouraged to investigate soft-decision quantization. Fu et al. [14] present a multi-bit quantization fusion scheme for CSS in which each SU employs *q*-bit quantization instead of a 1-bit hard-decision, compares the energy value during a sensing interval with a pair of quantization thresholds, and then produces multi-bit data. The FC collects quantized multi-bit data from all the SUs and performs inverse quantization, then softly combines the data. Fan et al. [15] introduce general log_2_*M*-bit local quantization, and jointly consider the diversity and SNR gains, where local sensing is divided into multiple regions in order to obtain the closed-form expressions for both the local thresholds and fusion rule. Chen et al. [16] employ *M*-ary quantized data to achieve spectrum sensing at the FC, and consider defense against spectrum sensing data falsification (SSDF) attacks. In [17], each SU does not make any decision, and only forwards the information it obtains via sensing to the FC through an imperfect reporting channel with a Nakagami-m distribution.

However, the reporting overhead grows as the number of quantization bits increases. In [18], the authors propose a multi-bit fusion rule where the SUs send their 1-bit sensing results to the FC instead of sending multi-bit quantization information, which results in multi-bit reporting and an increase in the energy efficiency. Bhowmick et al. [19] discuss a hybrid fusion scheme for CSS in presence of fading effects, in which the SUs send a hard-decision if the quality of the reporting channel is sufficient; otherwise, they send a soft-decision. Received hard and soft-decisions are combined via the OR rule at the FC to obtain the final decision. In [20], the optimization problem is about maximizing the network throughput subject to a target detection probability that is constructed. The authors optimize the number of sensing samples and the number of reporting bits for quantization, thereby obtaining the optimal quantized reporting bits for a given detection performance. Yucek et al. [21] propose a semi-soft method that illustrates the tradeoff between the sensing performance and local overheads. Here, the FC estimates the global test statistics and makes the final decision. However, the global fusion is not precise due to data estimation based on the mean value.

Motivated by the above discussion, this paper proposes a novel semi-soft decision scheme for CSS in CR networks in order to achieve a sufficient tradeoff between the sensing performance and band cost. Each SU performs local sensing by the energy detection (ED) method [22,23] and sends decision results to the FC. The FC collects all the sensed information and estimates the received decision data through a data reconstruction module based on a statistical distribution. At last, the global test statistic is computed by the FC to decide whether the PU signal is present or not. The major contributions of this paper can be summarized as follows:
(1)Most prior studies achieve fine sensing performance by means of multi-bit quantization fusion. The more quantization bits, the better the sensing performance, which ignores other constraints in CSS, such as the transmission bandwidth. However, in an actual communication system, these factors must be taken into account together. Through this, the proposed decision scheme designs a novel strategy through the reconstruction of a global test statistic to achieve a sufficient tradeoff between the sensing performance and the transmission bandwidth. The closed-form expression of the average transmission bandwidth is also derived.(2)In the prior studies, the data estimation at the FC is based on extra thresholds as well as the mean value, which results in an inaccurate estimation that affects the sensing performance. To address this issue, our proposed scheme is based on a semi-soft fusion rule in which three thresholds are required for the local decision. In the range where the data is not easily misjudged, we employ a method, through a statistical distribution of the local sensing data (truncated normal distribution), to estimate the measuring data of each SU. In the middle range, the mean value method can be used to estimate the data under a certain error tolerance.(3)Via MATLAB simulations, the superiority of the proposed method is further verified compared with existing algorithms in various situations. Overall, we find that the detection performance of the proposed scheme is only inferior to soft-decision detection, but makes savings on the transmission bandwidth. In addition, we provide a detailed analysis of the effects of various parameters on the performance of the proposed scheme. With an increase in the number of SUs, the number of samples, and the parameter α, the probability of detection and the average global test statistics can be improved accordingly.


The rest of this paper is organized as follows. In Section 2, the system model is formulated. The proposed semi-soft decision scheme is introduced in Section 3. Results from the simulation of various algorithms are provided in Section 4. Section 5 concludes the paper.

## 2. The System Model

We consider a CSS network that consists of a PU, an FC, and *M* SUs as depicted in Figure 1, where *M* SUs collaboratively detect whether the PU signal exists or not. Each SU employs the ED method for local spectrum sensing and determines the transmitted data. Hypothesis 0 (*H*_0_) denotes the absence of the PU; hypothesis 1 (*H*_1_) denotes that the PU is active and the channel is busy. The FC collects all the transmitted data and reconstructs the global test statistics, then makes a final decision by the decision rule.

The local sensing of SUs plays a key role in the above system as the first step to achieve spectrum sensing. During the local spectrum sensing process, the received signal of each SU can be formulated as
(1)$$\begin{array}{cc} x_{i}(n)\;=w_{i}(n), & H_{0}\\ x_{i}(n)\;=h_{i}s(n)+w_{i}(n), & H_{1} \end{array}$$
where *x_i_*(*n*) represents the received signal at the *n*-th moment by the *i*-th SU, *h_i_* denotes the channel gain between the PU and the *i*-th SU, *s*(*n*) is the transmitted signal of PU, and *w_i_*(*n*) is a zero-mean additive white Gaussian noise (AWGN) with a variance $\sigma_{w}^{2}$. Without loss of generality, we also assume that *s*(*n*) and *w_i_*(*n*) are independent of each other. Then, the *i*-th SU uses the energy of the received samples as the test statistics for the energy detector, which is given by
(2)$$X=\sum_{n=1}^{N}{\left|x_{i}(n)\right|}^{2}$$

Thus, *X* follows a central chi-square distribution with *N* degrees of freedom under *H*_0_, and follows a non-central chi-square distribution under *H*_1_. According to the Central Limit Theorem (CLT) [24], when the number of samples *N* is large enough, the distribution of *X* can be approximated as a Gaussian distribution [25], that is,
(3)$$\left\{ \begin{array}{cc} X\sim N(N\sigma_{w}^{2},2N\sigma_{w}^{4}) & H_{0}\\ X\sim N(N(1+\gamma )\sigma_{w}^{2},2N{\left(1+\gamma \right)}^{2}\sigma_{w}^{4}) & H_{1} \end{array}\right.$$
where *N*(*μ*,$\sigma_{w}^{2}$) denotes Gaussian distribution with mean *μ* and variance $\sigma_{w}^{2}$
*γ* is the SNR.

Therefore, the probability of a false alarm *P_f_* and the probability of detection *P_d_* are respectively calculated as:(4)$$P_{f}=\mathrm{Pr}(X\geq \lambda |H_{0})=Q\left(\frac{\lambda -N\sigma_{w}^{2}}{\sqrt{2N\sigma_{w}^{4}}}\right)$$
(5)$$P_{d}=\mathrm{Pr}(X\geq \lambda |H_{1})=Q\left(\frac{\lambda -N(1+\gamma )\sigma_{w}^{2}}{\sqrt{2N{\left(1+\gamma \right)}^{2}\sigma_{w}^{4}}}\right)$$
where the decision threshold *λ* can be expressed as
(6)$$\lambda =\sigma_{w}^{2}\left(Q^{-1}\left(P_{f}\right)\sqrt{2N}+N\right)$$

Here, *Q*(*x*) is given as follows:(7)$$Q\left(x\right)=\frac{1}{\sqrt{2\pi }}\int_{x}^{\infty }\exp(-\frac{t^{2}}{2})dt$$

## 3. The Proposed Semi-Soft Decision Scheme

In our proposed scheme, the sensing strategy is designed based on three modules: the local detection module, the data reconstruction module, and the global decision module. The local detection module is used to detect the PU signal by ED and send the results of the decision in terms of 1-bit or 2-bit information. The data reconstruction module is responsible for estimating the received decision data. The global decision module is in charge of fusing the estimated data and making a final decision.

### 3.1. Local Detection Module

In order to achieve a better tradeoff between the sensing performance and the local bandwidth cost, the proposed scheme is based on a semi-soft fusion rule in which three thresholds (i.e., λ, λ_1_, λ_2_) are required for the local decision, as shown in Figure 2.

According to Equation (6), the other two decision thresholds λ_1_ and λ_2_ can be obtained via
(8)$$\begin{array}{l} \lambda_{1}=(1+\alpha )\lambda \\ \lambda_{2}=(1-\alpha )\lambda \end{array}$$
where *α* is a given parameter. Obviously, *λ*_1_ and *λ*_2_ are subject to parameters *λ* and *α*.

Each SU independently makes the decision according to a comparison between the test statistics and the thresholds, as shown in Table 1.

### 3.2. Data Reconstruction Module

It is well known that the sensing performance is limited by the number of local quantization bits in CSS. In order to further improve the sensing performance, we reconstruct the transmitted data from each SU based on the statistical distribution, and the FC estimates the received decision data through this module. Therefore, the data reconstruction module is an essential part of the proposed scheme.

The received signal of SU follows a Gaussian distribution with mean *μ*_0_ and variance *σ*2 0 under *H*_0_ and follows a Gaussian distribution with mean *μ*_1_ and variance *σ*2 1 when the channel is busy. By bonding the random variables with the thresholds, the normal distribution can be derived from the truncated normal distribution, so we can estimate the data in the FC based on the theoretical distribution of the random variables.

As illustrated in Figure 3, when the local decision is *X_i_* < *λ*_2_ or *X_i_* > *λ*_1_, the local test statistics are far away from the threshold λ. The decision results are often accurate; therefore, we can utilize the corresponding truncated normal distribution to reconstruct the measuring data of each SU. In the range where the decision is close to the threshold *λ*, that is *λ*_2_ ≤ *λ* ≤ *λ*_1_, the decision results tend to be misjudged. Under a certain error tolerance, the mean value method can be used to estimate the data.

Therefore, the data estimation can be fulfilled as
(9)$$X_{i}^{fc}=\left\{ \begin{array}{cc} \mu_{0}-\sigma_{0}\frac{\phi (\frac{\lambda_{2}-\mu_{0}}{\sigma_{0}})}{\Phi (\frac{\lambda_{2}-\mu_{0}}{\sigma_{0}})} & 0\\ \frac{\lambda_{2}+\lambda }{2} & 00\\ \frac{\lambda +\lambda_{1}}{2} & 11\\ \mu_{1}+\sigma_{1}\frac{\phi (\frac{\lambda_{1}-\mu_{1}}{\sigma_{1}})}{1-\Phi (\frac{\lambda_{1}-\mu_{1}}{\sigma_{1}})} & 1 \end{array}\right.$$
where *Φ* (·) denotes the probability density function (PDF) of the standard normal distribution with zero mean and unit variance, that is,
(10)$$\phi (x)=\frac{1}{\sqrt{2\pi }}\exp(-\frac{1}{2}x^{2}).$$

Φ(·) is the cumulative distribution function (CDF) of the standard normal distribution, that is,
(11)$$\Phi (x)=\int_{-\infty }^{x}\frac{1}{\sqrt{2\pi }}\exp(-\frac{1}{2}t^{2})dt.$$

### 3.3. Global Decision Module

After the data estimation, all the reformulated data are fused according to the fusion rule and a final decision is made. Although the global decision module is easy to implement, it plays a decisive role in the scheme. The global test statistic *T_fc_* is formulated as follows:(12)$$T_{fc}=\sum_{i=1}^{M}\omega_{i}X_{i}^{fc}$$
where *ω**_i_* is the given weight to each estimated test statistic, *ω**_1_ +*
*ω**_2_ +···+ ω**_M_ =* 1. The FC employs the equal gain combining (EGC) diversity technique to calculate the global test statistic. Since the received signal of SU follows a Gaussian distribution, the sum of *M* Gaussian distributions can be still approximated as a Gaussian distribution, that is
(13)$$\left\{ \begin{array}{cc} X\sim N(MN\sigma_{w}^{2},2MN\sigma_{w}^{4}) & H_{0}\\ X\sim N(MN(1+\gamma )\sigma_{w}^{2},2MN{\left(1+\gamma \right)}^{2}\sigma_{w}^{4}) & H_{1} \end{array}\right..$$

Subsequently, the expression of the global probability of false alarm *Q_f_* and the global probability of detection *Q_d_* can be respectively given as follows:(14)$$Q_{f}=Q\left(\frac{\lambda_{fc}-MN\sigma_{w}^{2}}{\sqrt{2MN\sigma_{w}^{4}}}\right)$$
(15)$$Q_{d}=Q\left(\frac{\lambda_{fc}-MN(1+\gamma )\sigma_{w}^{2}}{\sqrt{2MN{\left(1+\gamma \right)}^{2}\sigma_{w}^{4}}}\right).$$

Applying the Neyman–Pearson criterion, we can obtain the expression of the global decision threshold *λ_fc_*, that is
(16)$$\lambda_{fc}=\sigma_{w}^{2}\left(Q^{-1}\left(Q_{f}\right)\sqrt{2MN}+MN\right).$$

Then, the final decision is made via comparing the global test statistic *T_fc_* with the threshold *λ**_fc_*. The decision rule can be expressed as
(17)$$\left\{ \begin{array}{cc} T_{fc}\geq \lambda_{fc} & H_{1}\\ T_{fc}<\lambda_{fc} & H_{0} \end{array}\right.$$

In summary, the proposed sensing method in the scheme is revealed as Algorithm 1. Each SU performs local sensing and sends the results to the FC, then the FC collects the information and estimates the received data so as to make the final decision.

**Algorithm 1.** The Proposed Semi-Soft Decision Scheme**Local detection module:**1. The PU broadcasts its signal and each SU calculates the test statistic *X* according to Equation (2). The decision threshold λ can be obtained according to Equation (6).2. Each SU performs local sensing according to Table 1 and sends the results (e.g., ’’0′’’’1′’’’00′’’’11′’) to the FC.**Data reconstruct****ion module****:**3. The FC collects all the sensing information of each SU.4. When the transmitted data is “0” or “1”, the FC reconstructs data based on the truncated normal distribution; otherwise, it adopts the uniform distribution to reconstruct data according to Equation (9).**Global decision module:**5. The global test statistic is fused at the FC according to Equation (12). The global decision threshold λ*_fc_* can be obtained according to Equation (16).6. The final decision is made according to Equation (17). If *T_fc_ ≥ λ_fc_*, the FC decides the PU signal is present, or the PU signal is absent.

In order to show the superiority of the proposed scheme, we first analyze the average local bandwidth cost and then demonstrate the detection performance in Section 4. The bandwidth of the proposed method is subject to the decision thresholds *λ*_1_ and *λ*_2_. Let $p_{i,d}^{1}$ and $p_{i,f}^{1}$ denote the probability of detection and false alarm, respectively, of the *i*-th SU corresponding to the threshold λ_1_. Similarly, $p_{i,d}^{2}$ and $p_{i,f}^{2}$ denote the probability of detection and false alarm, respectively, corresponding to the threshold λ_2_.
(18)$$P_{i,f}^{1}=Q\left(\frac{\lambda_{1}-N\sigma_{w}^{2}}{\sqrt{2N\sigma_{w}^{4}}}\right)\;P_{i,d}^{1}=Q\left(\frac{\lambda_{1}-(1+\gamma )N\sigma_{w}^{2}}{\sqrt{2N(1+\gamma {)}^{2}\sigma_{w}^{4}}}\right)\;P_{i,f}^{2}=Q\left(\frac{\lambda_{2}-N\sigma_{w}^{2}}{\sqrt{2N\sigma_{w}^{4}}}\right)\;P_{i,d}^{2}=Q\left(\frac{\lambda_{2}-(1+\gamma )N\sigma_{w}^{2}}{\sqrt{2N(1+\gamma {)}^{2}\sigma_{w}^{4}}}\right)$$

In *H*_0_ and *H*_1_, the probabilities that denote the test statistic of the *i*-th SU lie between the thresholds *λ*_1_ and *λ*_2_, respectively, expressed as follows:(19)$$P_{0}=P_{i,f}^{2}-P_{i,f}^{1}\;P_{1}=P_{i,d}^{2}-P_{i,d}^{1}$$

Therefore, the probability of the *i*-th SU transmitting 2-bit data can be calculated as
(20)$$\begin{array}{ll} P_{b}^{i} & =P(H_{0})P_{0}+P(H_{1})P_{1}\\ & =P(H_{0})(P_{i,f}^{2}-P_{i,f}^{1})+P(H_{1})(P_{i,d}^{2}-P_{i,d}^{1}) \end{array}$$

Subsequently, the average transmission bandwidth can be given as follows:(21)$$B=\frac{1}{M}\sum_{i=1}^{M}2P_{b}^{i}+(1-P_{b}^{i})$$

## 4. Simulation Results

In this section, the performances of the proposed method are illustrated via 10^4^ Monte Carlo simulations in MATLAB by comparison with soft-decision, hard-decision (OR rule), and semi-soft decision in [21]. The PU signal and noise are zero-mean with variance 1. In the following simulations, we analyze different situations, such as the SNR, the number of SUs, and the number of samples on the detection performance of the proposed method.

Figure 4 compares the probability of detection *P_d_* against different SNRs for each decision. When the probability of false alarm *P_f_* is set as 0.01, we find that soft-decision has the best performance, hard-decision has the worst performance, and the performance of two other decision algorithms lies between that of soft and hard-decision. It is worth mentioning that the performance of the proposed method is close to that of soft-decision. Provided that the sensing performance needs to meet the requirements of the IEEE 802.22 standard [26], when *P_d_* = 0.9, the SNR gain required by the proposed method is about 0.2 dB higher than that of soft-decision, about 1 dB lower than that of semi-soft decision, and about 2.4 dB lower than that of hard-decision.

Figure 5 presents the receiver operating characteristics (ROCs) of each method, which further demonstrates the advantage of the proposed method. As can be seen in the figure, the sensing performance of the proposed method is close to that of soft-decision, and is far superior to that of hard-decision. When the probability of false alarm *P_f_* is set as 0.1, the probability of detection *P**_d_* of soft-decision can achieve 0.82 and the proposed method can achieve 0.78, while the semi-soft decision scheme in [21] and hard-decision only achieve 0.59 and 0.48, respectively.

Figure 6a,b show the probability of detection *P_d_* curves of the proposed method for different numbers of SUs and the number of samples *N*. The probability of false alarm *P_f_* = 0.01. From Figure 6a, it can be observed that, as the number of SUs increases from 5 to 30, the probability of detection *P_d_* is improved accordingly. For a given SNR = −16 dB, the probability of detection *P_d_* could reach about 0.15, 0.3, 0.53, and 0.72, respectively. From Figure 6b, it can be seen that the probability of detection *P_d_* is also enhanced when the number of samples *N* takes a value from 500 to 2000. When *N* = 500 and SNR = −9 dB, the probability of detection *P_d_* can reach 1. Therefore, the detection performance can be improved by increasing of the number of SUs or the number of samples when the SNR is low.

Figure 7 shows the probability of detection *P_d_* curves of the proposed method with a different given parameter α for fixed *P_f_* = 0.01. We can find that the detection performance can be improved clearly by increasing α from 0.02 to 0.1. Meanwhile, there is a wider range between the decision thresholds λ_1_ and λ_2_, which means that a higher number of SUs will send 2-bit information data to the FC, and that more transmission bandwidth may be needed accordingly.

The reconstruction of the global test statistics of the proposed method is described in Figure 8. As is shown, the average global test statistics grow with the increase in the number of SUs. Besides this, the estimated test statistics by the proposed method are close to the average test statistics by soft-decision and significantly exceed those by semi-soft decision, which demonstrates the accuracy of the reconstruction of the test statistics by the proposed method.

## 5. Conclusions

In this paper, we have proposed a novel semi-soft decision scheme for CSS in CR networks in order to achieve a fulfilling tradeoff between the sensing performance and band cost. Each SU performs a local sensing process based on three thresholds and sends the results of its decision in terms of 1-bit or 2-bit data to the FC. Subsequently, we utilize the statistical distribution of the local sensing data to estimate the received decision data of each SU in order to obtain an accurate estimation. Finally, the global test statistic is fused by the FC to decide whether the PU signal is present or not. Results from a simulation show that the detection performance of the proposed method is only inferior to that of soft-decision, but makes savings on the transmission bandwidth. In addition, we provide a detailed analysis of the impact of various parameters on the performance. With an increase in the number of SUs, the number of samples, and the parameter α, the probability of detection and the average global test statistics can be improved accordingly. A closed-form expression of the average transmission bandwidth was also derived.

## Figures and Tables

**Figure 1 sensors-19-02522-f001:**
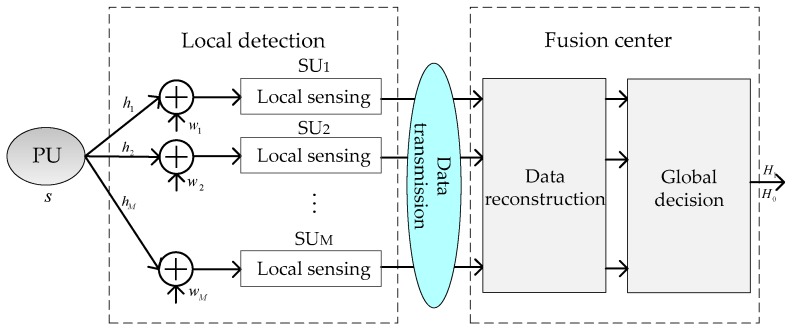
The structure of the cooperative spectrum sensing (CSS) system with the novel semi-soft decision rule.

**Figure 2 sensors-19-02522-f002:**
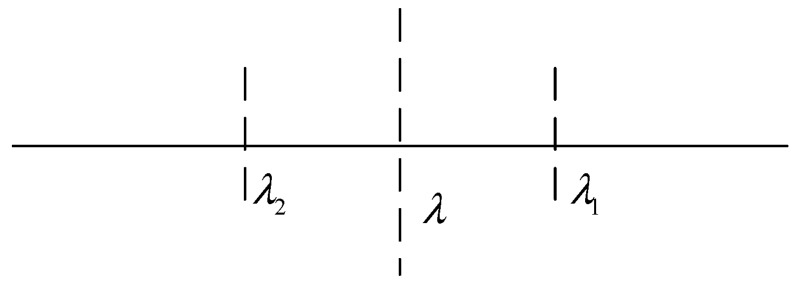
The local decision thresholds.

**Figure 3 sensors-19-02522-f003:**
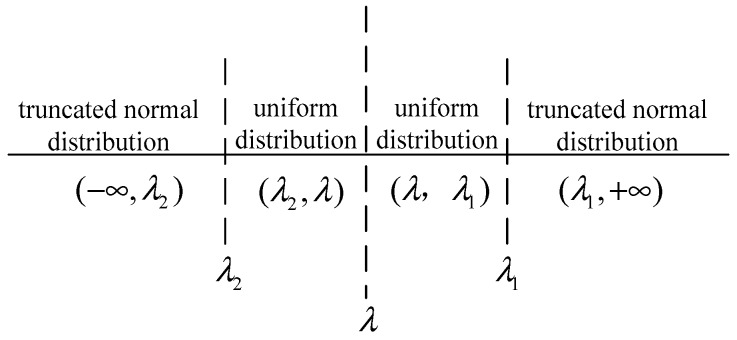
The distribution of the reconstructed data.

**Figure 4 sensors-19-02522-f004:**
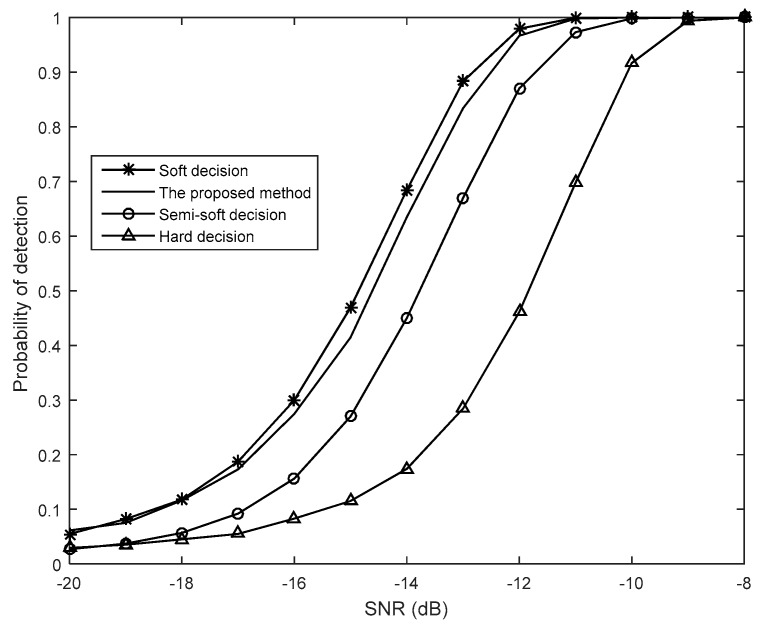
Comparisons of the probability of detection *P_d_* (*M* = 10, *N* = 1000, *P_f_* = 0.01).

**Figure 5 sensors-19-02522-f005:**
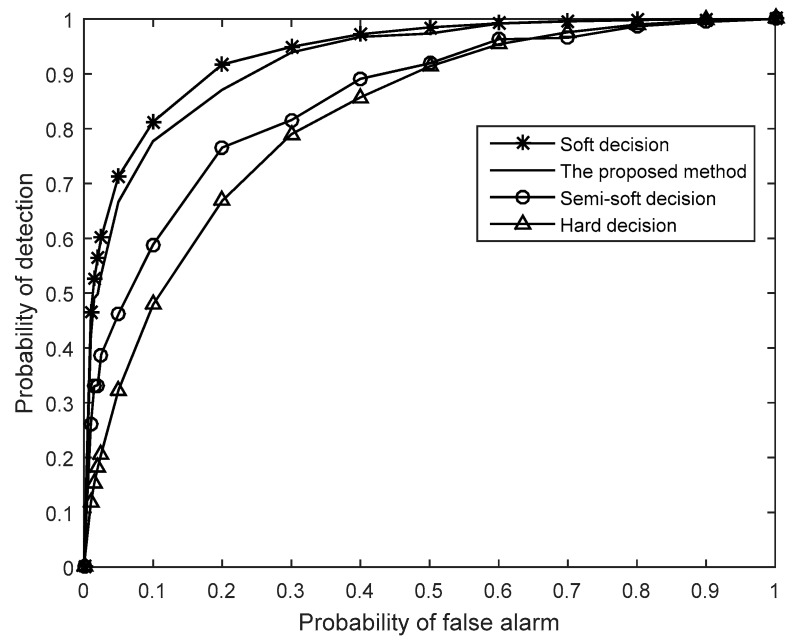
Comparisons of the receiver operating characteristic (ROC) curves (*M* = 10, *N* = 1000, SNR = −15 dB).

**Figure 6 sensors-19-02522-f006:**
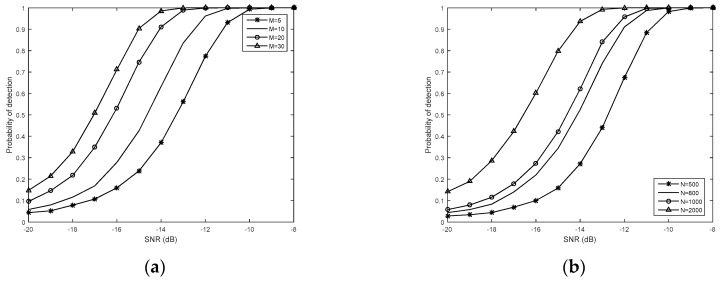
The probability of detection *P_d_* curves of the proposed method: (**a**) Different number of secondary users (SUs) (*M* = 5, 10, 20, 30, *N* = 1000); (**b**) Different numbers of samples (*N* = 500, 800, 1000, 2000, *M* = 10).

**Figure 7 sensors-19-02522-f007:**
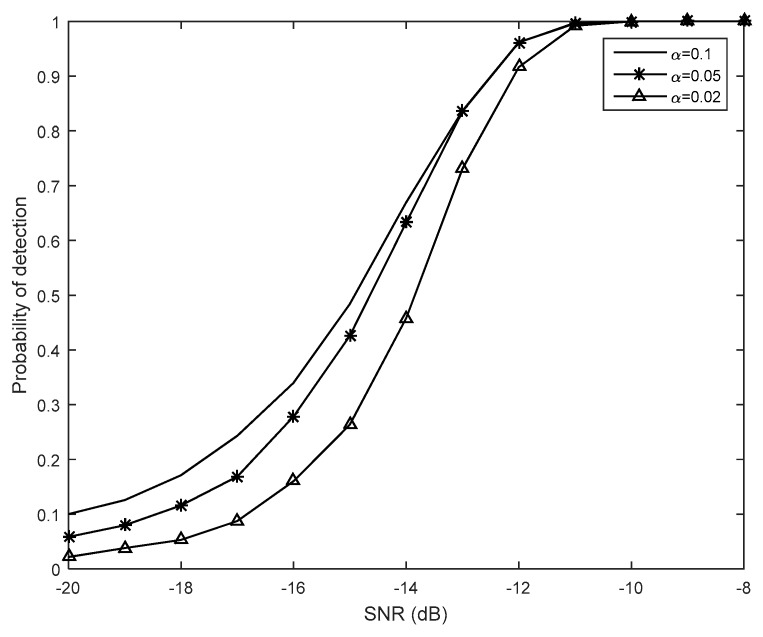
The probability of detection *P_d_* curves with a different given parameter α (*M* = 10, *N* = 1000, *P_f_* = 0.01).

**Figure 8 sensors-19-02522-f008:**
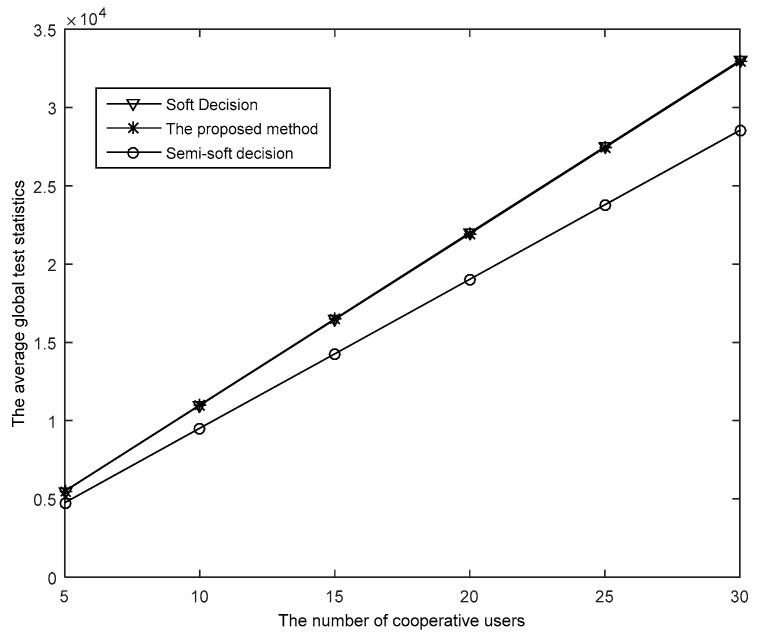
The relationship between the number of SUs and the average global test statistics (α = 0.05, *N* = 1000, *P_f_* = 0.01).

**Table 1 sensors-19-02522-t001:** The transmitted data and data size of the local decision.

The Local Decision	Transmitted Data	Data Size
$X_{i}<\lambda_{2}$	0	1 bit
$X_{i}>\lambda_{1}$	1	1 bit
$\lambda_{2}\leq X_{i}<\lambda$	00	2 bits
$\lambda <X_{i}\leq \lambda_{1}$	11	2 bits

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
