# Peer review of "A Novel Semi-Soft Decision Scheme for Cooperative Spectrum Sensing in Cognitive Radio Networks"

_sensors, 2019, doi:10.3390/s19112522_

Round 1
Reviewer 1 Report
The paper proposes a new semi-soft decision scheme for collaborative spectrum sensing. According to this scheme each SU first makes local energy detection decisions according to three thresholds (dependent on the desired false alarm detection probability) and sends the decisions to a fusion center for making the final decision. The scheme is evaluated via Monte-Carlo simulation, comparing against soft, hard and another semi-soft decision-making approaches.
The topic is modern and can have potential interest for theory and practical applications. It could of interest for the audience of this periodical.
The novelty is restricted given the broader body of existing knowledge on collaborative spectrum sensing.
The decision-making in the fusion center has not been explained adequately. A more elaborate and clear description is needed.
Linguistic revision is needed, as there are various typos remaining across the manuscript. Some parts of the text require revision, some of which are lines 20-21, 28-31 and 211-213.
The caption of figure 5 should be right below it, rather than in new page.
Author Response
Point 1: The novelty is restricted given the broader body of existing knowledge on collaborative spectrum sensing.
Response 1: Thank you very much for this comment. As you said, there are many existing researches on collaborative spectrum sensing.
Some methods apply hard decision rule to fuse the information, but the better detection performance can not be obtained. Some schemes concentrate on the enhancement of the detection performance by means of multi-bit quantization fusion. The more quantization bits, the better sensing performance, which ignore other constraints in CSS such as the transmission bandwidth. However, in an actual communication system, these factors have to be taken into account together. So, the proposed decision scheme designs a novel strategy through the reconstruction of global test statistic to achieve a good tradeoff between the sensing performance and the transmission bandwidth, in addition, the closed-form expression of the average transmission bandwidth is also derived.
Point 2: The decision-making in the fusion center has not been explained adequately. A more elaborate and clear description is needed.
Response 2: Thank you for this comment. We have added an elaborate description about the global decision process in the revised manuscript as follows.
The FC employs equal gain combining (EGC) diversity technique to calculate the global test statistic, since the received signal of SU follows Gaussian distribution, the sum of M Gaussian distributions can be still approximated as Gaussian distribution, that is
(13) |
Subsequently, the expression of the global probability of false alarm Qf and the global probability of detection Qd can be respectively given as follows:
(14) | |
(15) |
Applying the Neyman-Pearson criterion, we can get the expression of the global decision threshold λfc , that is
(16) |
The corresponding revisions have been highlighted in page 6 of the revised manuscript.
Point 3:Linguistic revision is needed, as there are various typos remaining across the manuscript. Some parts of the text require revision, some of which are lines 20-21, 28-31 and 211-213.
Response 3: Thank you very much for this comment. In our manuscript, lines 20-21 are revised to ”Secondly, and most importantly, the FC estimates the received decision data through data reconstructing module based on the statistical distribution that the extra thresholds are not needed.”;
Lines 28-31 are revised to “With the rapid development of Internet of Things (IoT) [1] and wireless applications, especially for the fifth generation (5G) communication systems and beyond [2], the spectrum resources are becoming increasingly strained. Currently, the radio spectrum is mainly used based on a fixed spectrum allocation policy. ”;
Lines 211-213 are revised to”In this section, the performances of the proposed method are illustrated via 104 Monte Carlo simulations in MATLAB by comparing with soft decision, hard decision (OR rule), and semi-soft decision in [21]. The PU signal and noise are zero-mean with variance 1. In the following simulations, by analyzing the effect of SNR, the number of SUs, and the number of samples on the detection performance to verify the superiority of the proposed method.”.
Also,we have revised the language of the entire manuscript. The corresponding revisions have been highlighted in red in the revised manuscript.
Point 4:The caption of figure 5 should be right below it, rather than in new page.
Response 4:Thank you for this comment. In the revised manuscript, we have made adjustment.

Reviewer 2 Report
The paper presents a tradeoff approach to the design of cooperative spectrum sensing in cognitive radio networks. The sensing performance and band cost are compromised in their design. The Matlab simulation has been carried out for performance study.
The organization and presentation of the theoretical work are acceptable. However, it is suggested to compare the proposed scheme with the semi-soft approach. That is, the distinctive advantages of the proposed scheme over the semi-soft one could be discussed in technical depth.
Author Response
Point 1:The organization and presentation of the theoretical work are acceptable. However, it is suggested to compare the proposed scheme with the semi-soft approach. That is, the distinctive advantages of the proposed scheme over the semi-soft one could be discussed in technical depth.
Response 1:Thank you very much for this comment. According the threshold division, when the test statistics lie the range where the data is not easily misjudged, that is Xi<λ2 or Xi>λ1, we employs a method through statistical distribution of the local sensing data, truncated normal distribution, to estimate the measuring data of each SU. While in [λ2,λ1], the mean value method can be used to estimate the data under a certain error tolerance. However, the semi-soft approach employs the mean value to estimate the data in whole threshold ranges.
What is clear is that the truncated normal estimation is an excellent unbiased estimation compared with the mean value in statistics. In many engineering fields, the example calculation shows that the reliability under the truncated distribution is better than the mean value method. Meanwhile, in our simulations, Figure 8 demonstrates an accuracy of reconstructing test statistics by the proposed method. In the future, we will perform an in-depth study of statistics.
Due to space limitations, an introduction to the truncated normal distribution can be described fully as follows:
[1] https://en.wikipedia.org/wiki/Truncated_normal_distribution
[2] Johnson, N.L., Kotz, S., Balakrishnan, N. (1994) Continuous Univariate Distributions, Volume 1, Wiley. ISBN 0-471-58495-9 (Section 10.1)
